# Integrative Genomic and Clinicopathologic Characterization of Pure Primary Ovarian Large Cell Neuroendocrine Carcinoma: A Case Report and Molecular Insight

**DOI:** 10.3390/curroncol32100540

**Published:** 2025-09-27

**Authors:** Hyonjee Yoon, Chaewon Kim, Yongseok Lee, Jimin Ahn, Minjin Jeong

**Affiliations:** Department of Obstetrics and Gynecology, Eunpyeong St. Mary’s Hospital, College of Medicine, The Catholic University of Korea, Seoul 03312, Republic of Korea; ob.jee.y@catholic.ac.kr (H.Y.); chaegy@catholic.ac.kr (C.K.); gom@catholic.ac.kr (Y.L.); jmahn9978@catholic.ac.kr (J.A.)

**Keywords:** high-grade neuroendocrine neoplasm, ovarian large cell neuroendocrine carcinoma, next-generation sequencing (NGS), precision oncology

## Abstract

Primary ovarian large cell neuroendocrine carcinoma (LCNEC) is a very rare and aggressive cancer, with no established treatment guidelines. We report a case in a postmenopausal woman treated with complete surgical resection followed by platinum-based chemotherapy. Comprehensive genomic profiling revealed a *BRCA2* mutation, homologous recombination deficiency (HRD), microsatellite instability-high (MSI-H), and a very high tumor mutational burden (TMB). These features resemble those seen in other gynecologic neuroendocrine carcinomas and suggest possible benefit from targeted therapies such as PARP inhibitors or immune checkpoint blockade. This case highlights the value of genomic analysis in rare gynecologic cancers to guide personalized treatment.

## 1. Introduction

Large cell neuroendocrine carcinoma (LCNEC) is a high-grade, poorly differentiated neuroendocrine tumor with aggressive behavior and a high mitotic rate [1]. According to the 2020 WHO classification, LCNEC is categorized as a high-grade neuroendocrine tumor with histologic overlap with small cell carcinoma but distinct morphological features [2]. While most frequently observed in the lung, LCNEC can arise in extrapulmonary sites, including the gastrointestinal tract and female genital tract. LCNEC has been reported in the cervix, endometrium, and ovary among gynecologic tumors, with the ovary being the least commonly affected site [3]. Since its first description in 1991, only about 25 cases of pure primary ovarian LCNEC have been reported [4]. Diagnosis is typically based on histopathology and immunohistochemistry, showing positivity for neuroendocrine markers such as synaptophysin and chromogranin [5]. Due to its rarity, optimal treatment strategies remain undefined. Reported outcomes suggest a high recurrence rate within the first year and a 5-year survival rate of 20–30% [6]. While prior case reports have focused primarily on histologic and clinical aspects, recent advances in molecular pathology have enabled a deeper understanding of tumor biology through genomic profiling.

Recent genomic profiling efforts have begun to elucidate the molecular heterogeneity of LCNEC and identify actionable targets. In pulmonary LCNEC, molecular profiling has identified distinct subtypes, such as small cell lung cancer (SCLC)-like variants characterized by concurrent *TP53* and *RB1* alterations and non-small cell lung cancer (NSCLC)-like variants that lack these co-alterations. This classification provides a framework for developing subtype-specific therapeutic strategies [7,8]. In contrast, genomic data on gynecologic LCNEC are extremely limited. To date, only a few pure ovarian LCNEC cases have undergone comprehensive genomic profiling, underscoring the urgent need for broader molecular characterization to inform precision-medicine strategies.

We report a rare case of pure primary ovarian LCNEC successfully treated with optimal debulking surgery followed by cisplatin-etoposide chemotherapy, incorporating next-generation sequencing (NGS) to explore potential therapeutic vulnerabilities.

## 2. Case Presentation

A 48-year-old nulliparous woman presented with a palpable pelvic mass and a three-month history of progressive abdominal discomfort. She had no notable medical or surgical history and no known family history of malignancy. Transvaginal ultrasound revealed a 16 cm multilocular cystic mass in the left adnexa with irregular echogenic solid components and papillary projections. Gynecologic sonography with Doppler imaging demonstrated internal vascularity within the solid areas, raising suspicion for malignancy (Figure 1a). No ascites or peritoneal implants were observed, and the contralateral adnexa appeared unremarkable. Pelvic MRI revealed that the mass exhibited uniformly low signal intensity on T1-weighted images, while T2-weighted sequences demonstrated a combination of low and high signal areas, indicative of a lesion with both cystic and solid components. (Figure 1b). Chest CT revealed no evidence of metastatic disease, and PET-CT showed only mild FDG uptake in a left para-aortic lymph node. Both upper and lower endoscopies were unremarkable. Serum tumor markers showed elevated CA-125 (68.5 U/mL), while HE4, CEA, CA19-9, and AFP were within normal limits. The ROMA score was low at 9.04% (premenopausal low-risk threshold: <11.4%). Exploratory laparotomy revealed a ruptured left ovarian mass measuring 16.0 cm × 12.0 cm × 9.0 cm, densely adherent to the sigmoid colon, rectum, and parietal pelvic peritoneum, without gross peritoneal implants or enlarged lymph nodes. The patient underwent total abdominal hysterectomy with bilateral salpingo-oophorectomy, pelvic and para-aortic lymphadenectomy, total omentectomy, and meticulous adhesiolysis.

Histopathologic examination confirmed a Grade 3 large cell neuroendocrine carcinoma (LCNEC) with capsular rupture and surface involvement. All 13 dissected lymph nodes were negative for metastasis. To differentiate this tumor from primary ovarian carcinomas, metastatic carcinomas, germ cell tumors, sex cord-stromal tumors, malignant melanoma, and leiomyosarcoma, an extended panel of immunohistochemical stains was performed (Figure 2). The tumor was positive for synaptophysin, CK-AE1/3, focal CDX-2, and focal chromogranin, while Müllerian markers (PAX8, WT-1, ER, and PR) were negative, supporting a diagnosis of primary ovarian LCNEC (Table 1). Although focal CDX-2 positivity raised suspicion for metastatic large cell neuroendocrine carcinoma of gastrointestinal origin [9], comprehensive systemic evaluation, including PET-CT and both upper and lower endoscopy, demonstrated no evidence of a gastrointestinal primary, effectively excluding this possibility. The patient received six cycles of cisplatin-etoposide chemotherapy (cisplatin 80 mg/m^2^ on day 1 and etoposide 80 mg/m^2^ on days 1–3). Follow-up included chest and pelvic CT at 3–6 month intervals with serial CA-125 evaluations, showing no evidence of recurrence over two years.

Postoperative next-generation sequencing (NGS) was performed on formalin-fixed, paraffin-embedded (FFPE) tumor tissue (ovarian mass; tumor cell content: 90%) using the Oncomine Comprehensive Assay Plus (Thermo Fisher Scientific, Waltham, MA, USA). Sequencing was conducted on the Ion S5 XL platform using the hg19 human genome reference. Data analysis and variant annotation were carried out using Torrent Suite v5.10.2, Ion Reporter v5.12, and Oncomine Knowledgebase Reporter v4.7. The assay targets approximately 500 cancer-related genes, including 165 hotspot genes, 227 full coding regions, 19 CNV gain-only genes, 51 fusion genes, and 86 TMB-only genes. Clinically significant pathogenic variants were identified in several genes. Most notably, a *BRCA2* frameshift mutation (c.7177dupA, p.M2393Nfs*19; variant allele frequency [VAF]: 41.75%) and an *ATM* nonsense mutation (c.1339C>T, p.R447; VAF: 42.85%) were detected, both categorized as Tier I variants. Additional Tier II mutations were found in *TP53*, *PTEN*, *APC*, *FANCD2*, *FANCM*, *RAD50*, *JAK2*, *FGFR2*, *PIK3R1*, *CTNNB1*, and *FBXW7*, suggesting disruptions in key oncogenic pathways such as DNA damage repair, Wnt/β-catenin, PI3K/AKT, and cell cycle regulation. No gene fusions or copy number alterations were identified. The detected variants are summarized in a gene-wise mutational panel (Figure 3), which illustrates variant allele frequencies together with corresponding amino acid and nucleotide changes for clarity. The tumor exhibited a markedly elevated tumor mutational burden (TMB) of 277.49 mutations/Mb and microsatellite instability-high (MSI-H) status with an MSI score of 34.92. Although the OCA Plus genomic integrity index could not be determined due to unavailable TAI and LST values, the presence of a pathogenic *BRCA2* mutation and MSI-H status strongly support homologous recombination deficiency (HRD). Peripheral blood germline *BRCA2* sequencing revealed no pathogenic alterations, confirming the somatic origin of the detected mutation.

These molecular features indicate potential sensitivity to PARP inhibitors and immune checkpoint inhibitors. However, clinical validation in ovarian LCNEC is still limited.

## 3. Discussion

Primary large cell neuroendocrine carcinoma (LCNEC) of the ovary is a highly aggressive and exceedingly rare neoplasm, with limited understanding due to its low incidence. It may present in either pure or mixed histologic forms and is frequently diagnosed at an advanced stage. Diagnostic confirmation relies on characteristic morphologic features supported by immunohistochemical staining, with synaptophysin and chromogranin being the most reliable neuroendocrine markers [12]. The immunohistochemical profile observed in our case is largely consistent with existing literature. Ovarian LCNECs typically express synaptophysin, chromogranin, and pan-cytokeratin (CK-AE1/3), while showing negative or variable staining for CK7, CDX-2, and EMA. Markers associated with Müllerian differentiation, such as PAX8, WT-1, ER, and PR, are characteristically negative, aiding in the differentiation from other high-grade epithelial ovarian carcinomas [12]. In our case, despite focal CDX-2 positivity, exhaustive clinical and radiologic evaluations ruled out a gastrointestinal primary, strongly supporting an ovarian origin. Additional negativity for CK20, GATA3, and glypican-3 further excluded metastatic gastrointestinal or germ cell tumors [13,14]. Clear cell carcinoma was also considered in the differential diagnosis. However, the tumor was negative for PAX8, WT-1, ER, and PR and showed only focal positivity for EMA, which is inconsistent with the typical immunophenotype of clear cell carcinoma. Furthermore, strong positivity for neuroendocrine markers synaptophysin and chromogranin reliably excluded clear cell carcinoma in this case.

Due to the absence of standardized guidelines, treatment of ovarian LCNEC is generally extrapolated from SCLC protocols [15]. Several case reports have documented favorable responses to cisplatin–etoposide chemotherapy in ovarian LCNEC, with some patients achieving prolonged remission [16,17]. Current NCCN guidelines for high-grade neuroendocrine carcinomas recommend platinum-based chemotherapy irrespective of primary site, and this remains the standard approach for ovarian LCNEC [18]. Other regimens, such as carboplatin combined with paclitaxel, especially in tumors with mixed epithelial components, have also been tried, but responses are variable and often less effective in pure LCNEC [19,20]. In select cases, adjuvant radiotherapy to the tumor bed has been administered to prevent local recurrence; however, systematic data on the radiosensitivity of pure ovarian LCNEC are lacking [21,22].

Despite multimodal treatment strategies, clinical outcomes remain poor, with short-lived responses and high recurrence rates. In this context, comprehensive molecular profiling has emerged as a promising tool for guiding personalized treatment. Recent multi-institutional genomic analyses of gynecologic NECs, particularly those of cervical and endometrial origin, have revealed recurrent alterations in critical oncogenic pathways, most notably those involved in DNA damage repair (e.g., *BRCA1/2*, *MSH2*, *ATM*), PI3K/AKT/mTOR, and MAPK signaling. For example, Eskander et al. conducted a genomic profiling study of high-grade neuroendocrine cervical carcinomas and identified frequent mutations in *PIK3CA* (19.6%), *MYC* (15.5%), *TP53* (15.5%), and *PTEN* (14.4%), and reported pathogenic alterations in the mismatch-repair gene *MSH2* (3%). A small subset of tumors (≈2%) demonstrated high tumor mutational burden (TMB-H) and microsatellite instability-high (MSI-H) status [23]. Similarly, Frumovitz et al. identified actionable mutations in *PIK3CA*, *KRAS*, and *TP53* in small cell neuroendocrine carcinoma of the cervix [24]. In endometrial NECs, genomic alterations involving *PIK3CA*, *PTEN*, *TP53*, *CTNNB1*, and *KRAS* have been reported, with overlapping molecular features also observed in epithelial endometrial carcinomas [25,26]. These findings suggest that, despite arising from distinct anatomical sites, high-grade NECs of the gynecologic tract may share a convergent molecular architecture, supporting the potential utility of biomarker-guided targeted therapies in this rare and aggressive cancer subtype.

While the molecular landscape of cervical and endometrial NECs has been increasingly delineated, ovarian LCNEC remains largely uncharacterized at the genomic level. To date, only one report has applied NGS to a case of pure primary ovarian LCNEC. In that study, Wang et al. reported a somatic *BRCA1* splice-site mutation, accompanied by HRD, MSI-high status, and elevated TMB. These molecular characteristics are similar to those observed in NECs of cervical and endometrial origin [27]. Similarly, in the largest clinicopathologic series to date, Flores Legarreta et al. reviewed 63 ovarian neuroendocrine neoplasms, including pure and mixed LCNEC as well as cases labeled small cell, high-grade, or neuroendocrine carcinoma not otherwise specified (NOS) [28]. In 26 patients (72%), 96% harbored ≥2 gene alterations; all 4 tested were PD-L1–positive, and 1 of 10 showed mismatch-repair deficiency.

Compared with prior reports, our case demonstrates unique distinctions. Wang et al. described a pure ovarian LCNEC harboring a somatic *BRCA1* splice-site mutation with HRD, MSI-H, and elevated TMB, whereas our patient exhibited a pathogenic *BRCA2* frameshift mutation accompanied by an *ATM* nonsense mutation. Similarly, while Flores Legarreta et al. reported frequent multigene alterations and immune-biomarker positivity in ovarian neuroendocrine neoplasms, the combined presence of *BRCA2* frameshift, *ATM* nonsense, MSI-H, and ultra-high TMB exceeding 200 mutations/Mb has not been documented to date.

Building on these distinctions, our case further expands the molecular landscape of ovarian LCNEC. We identified a pathogenic somatic *BRCA2* frameshift mutation (c.7177dupA), an *ATM* nonsense mutation, and additional Tier II alterations in *TP53* and *PTEN*. These were accompanied by HRD positivity, MSI-H status, and an exceptionally high TMB of 277.49 mutations/Mb, which represent molecular hallmarks closely aligned with those reported in previous studies of gynecologic NECs. This genomic convergence underscores the clinical relevance of comprehensive molecular profiling in rare tumor subtypes such as ovarian LCNEC, offering not only diagnostic clarity but also the opportunity to identify patients who may benefit from biomarker-guided therapeutic strategies.

The tumor in our case exhibited an extraordinarily high TMB of 277 mutations/Mb. This finding is likely explained by a combination of biological and technical factors. Biologically, the coexistence of MSI-H and deleterious DNA damage response (DDR) pathway mutations (*BRCA2* and *ATM*) may drive a hypermutated phenotype, as reported in subsets of colorectal and endometrial cancers. From a technical perspective, panel-based assays such as the Oncomine Comprehensive Assay Plus can yield variable estimates depending on panel size, content, and filtering. Current harmonization efforts emphasize that assay-specific cutoffs should be considered. Together, these observations support the validity of the observed ultra-high TMB while underscoring the importance of interpreting values in the context of assay limitations [29,30].

Based on the HRD, MSI-H, and TMB-H profiles, we discussed the potential use of PARP inhibitors or immune checkpoint inhibitors with the patient. Because of the rarity of pure ovarian LCNEC and the lack of prospective evidence, no maintenance therapy was initiated, and observation was chosen after platinum–etoposide. These agents remain rational options in recurrent or treatment-resistant disease, ideally within clinical trial settings. Although our patient ultimately did not receive PARP inhibitors or immune checkpoint inhibitors, the coexistence of HRD, MSI-H, and ultra-high TMB provides a compelling rationale for evaluating such targeted therapies in future trials. The parallels observed with other gynecologic NECs further support a unified molecular taxonomy and reinforce the need for site-agnostic, biomarker-guided treatment strategies. Conducting prospective clinical trials in such rare entities is inherently challenging because of extremely low incidence and biological heterogeneity. Collaborative international registries and inclusion in molecularly stratified basket or platform trials (e.g., NCI-MATCH, TAPUR-like designs) will therefore be critical to generate robust evidence and enable real-world data collection. International centralized registries are essential for rare gynecologic NECs, and participation in molecularly stratified basket trials may facilitate effective therapies. Nevertheless, this study is inherently limited by its single-case nature and the absence of functional validation for the identified genomic alterations. Despite these limitations, the comprehensive integration of histopathologic, immunohistochemical, and genomic data in this report provides novel insights into the biology of ovarian LCNEC. This case may serve as a meaningful reference for future research and the development of individualized treatment strategies.

## 4. Conclusions

Primary ovarian large cell neuroendocrine carcinoma is an exceptionally rare and aggressive malignancy, for which no standardized treatment guidelines currently exist. Our case highlights the critical role of accurate histopathologic diagnosis, complete cytoreductive surgery, and platinum-based chemotherapy as key therapeutic strategies. Our case adds to the evolving understanding of this malignancy by identifying clinically relevant genomic alterations, including *BRCA2* and *ATM* mutations, HRD, MSI-H, and ultra-high TMB. These findings highlight the potential role of genomic profiling in guiding precision therapy and underscore the need for biomarker-driven clinical research in this understudied entity.

## Figures and Tables

**Figure 1 curroncol-32-00540-f001:**
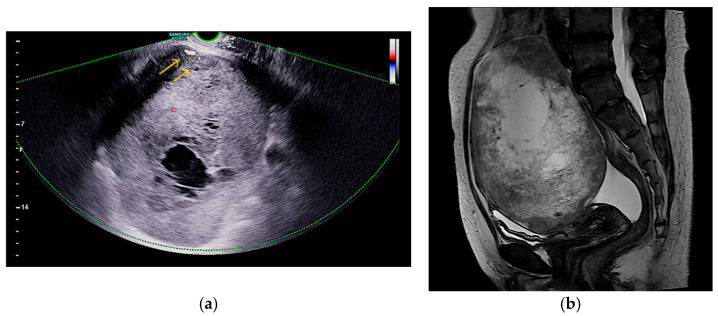
Imaging features of the pelvic mass: (**a**) Doppler ultrasound shows internal vascularity in solid areas without ascites or peritoneal implants, as indicated by the arrows; (**b**) Sagittal T2-weighted MRI shows mixed signal intensity, indicating cystic and solid components.

**Figure 2 curroncol-32-00540-f002:**
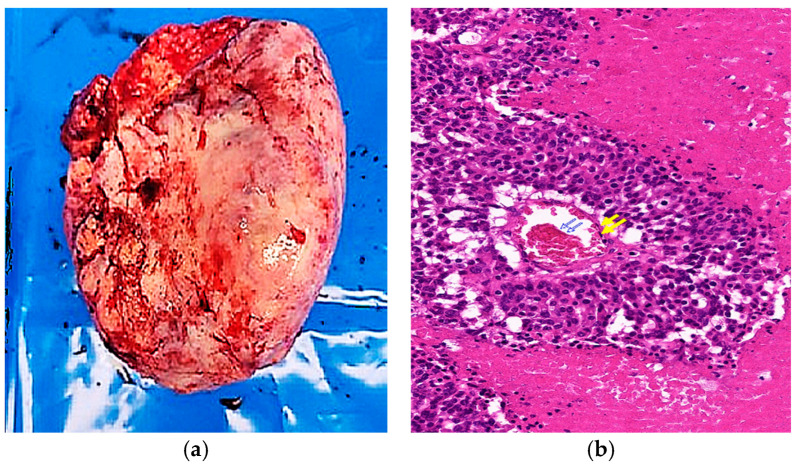
Gross, histologic, and immunohistochemical features of the tumor. (**a**) Macroscopic view showing an irregular outer surface with hemorrhagic areas and capsular rupture; (**b**) Low-power microscopy (×40, H&E) reveals nest architecture with central coagulative necrosis (blue arrows) and peripheral hemorrhage (yellow arrows); (**c**) High-power microscopy (×200, H&E) showing rosette-like structures (yellow arrows) and tumor cells with round-to-oval, hyperchromatic nuclei (red arrows); (**d**) Immunohistochemistry shows diffuse positivity for synaptophysin (×40); (**e**) focal positivity for chromogranin (×40); (**f**) diffuse cytoplasmic positivity for CK-AE1/3 (×40).

**Figure 3 curroncol-32-00540-f003:**
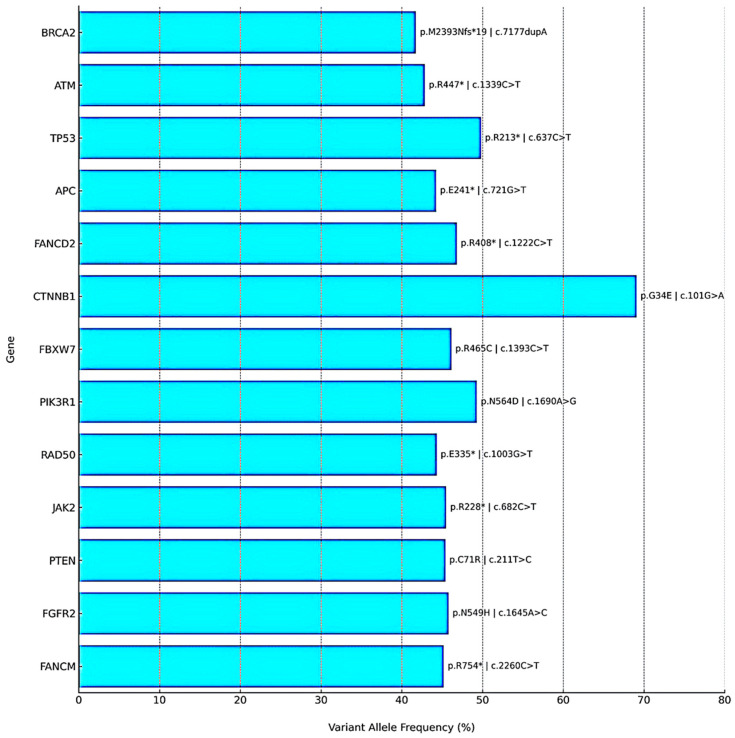
Gene-wise mutational panel detected by NGS. Bar plot of variant allele frequencies (VAF, %) with annotated amino acid and nucleotide changes. Tier I variants (*BRCA2*, *ATM*) and Tier II variants (e.g., *TP53*, *PTEN*, *APC*, *CTNNB1*) are shown. The tumor also demonstrated HRD positivity, MSI-H status, and ultra-high TMB. The asterisk (*) denotes a stop codon (nonsense mutation).

**Table 1 curroncol-32-00540-t001:** Immunohistochemical profile of the present case compared typical ovarian LCNEC findings reported in previous studies. (Reference data adapted from Gupta et al. 2021 [10] and Tsuji et al. 2019 [11]).

Marker	This Case (Present Study)	Typical Ovarian LCNEC (Literature)
CK-AE1/3	Positive	Positive
CK7	^1^ Focal positive	^2^ Variable
CK20	Negative	Negative
CDX-2	Positive	Variable
EMA	Focal positive	Variable
PAX8	Negative	Usually negative
WT-1	Negative	Negative
ER	Negative	Negative
PR	Negative	Negative
Synaptophysin	Positive	Positive
Chromogranin	Focal positive	Positive
CD117	Focal positive	Variable
CD30	Negative	Negative
Desmin	Negative	Negative
Calretinin	Negative	Negative
Vimentin	Negative	Negative
GATA3	Negative	Negative
AFP	Negative	Negative
Glypican-3	Negative	Negative
P63	Negative	Negative
HMB-45	Negative	Negative
HSA	Negative	Negative
P16	Focal positive	Variable
P53	Negative	Mutated/Overexpressed

^1^ Focal; Staining observed in a limited proportion of tumor cells. ^2^ Variable; Expression reported variably across published cases.

## Data Availability

The data presented in this study are available on request from the corresponding author.

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
