# Peer review of "Integrative Genomic and Clinicopathologic Characterization of Pure Primary Ovarian Large Cell Neuroendocrine Carcinoma: A Case Report and Molecular Insight"

_curroncol, 2025, doi:10.3390/curroncol32100540_

Round 1

Reviewer 1 Report

Comments and Suggestions for Authors

Interesting case and immunohistochemistry together with molecular analysis are adding novelty. The molecular findings are important for future therapeutic strategies and clinical trials. I suggest that the authors perform markers to exclude a clear cell carcinoma as from the pictures, this entity should be into differential diagnosis. Figure 2 b does not present with low power, necrosis and haemorhage and i suggest to delete it and replace it with a better one.

Author Response

Comments 1: I suggest that the authors perform markers to exclude a clear cell carcinoma as from the pictures, this entity should be into differential diagnosis.

Response 1: We appreciate the reviewer’s valuable comment. Although we did not perform HNF1β or Napsin A, which are representative markers for clear cell carcinoma (CCC), we would like to emphasize that the immunoprofile in this case is not compatible with CCC. The tumor was negative for PAX8, WT-1, ER, and PR, which contrasts with the characteristic positivity usually observed in CCC. Moreover, strong positivity for neuroendocrine markers (synaptophysin and chromogranin) confirmed the diagnosis of large cell neuroendocrine carcinoma (LCNEC). EMA was only focally positive, differing from the diffuse positivity often seen in CCC. Taken together, the immunohistochemical findings reliably exclude CCC, and we believe that additional markers are not necessary. We have clarified this point in the revised manuscript (revised version; page 7, line 191-196).

“Clear cell carcinoma was also considered in the differential diagnosis. However, the tumor was negative for PAX8, WT-1, ER, and PR, and showed only focal positivity for EMA, which is inconsistent with the typical immunophenotype of clear cell carcinoma. Furthermore, strong positivity for neuroendocrine markers synaptophysin and chromogranin reliably excluded clear cell carcinoma in this case.”

Comments 2: Figure 2b does not present with low power, necrosis and haemorhage and i suggest to delete it and replace it with a better one.

Response 2: We thank the reviewer for this valuable comment. The inappropriate image has been removed and replaced with a more suitable one. Key pathological findings have been highlighted with arrows and described in the figure legend accordingly (revised version; page 4, line 127-132).

Figure 2. (b) Low-power microscopy (×40, H&E) reveals nest architecture with central coagulative necrosis (blue arrows) and peripheral hemorrhage (yellow arrows).

(c) High-power microscopy (×200, H&E) shows rosette-like structures (yellow arrows) and tumor cells with round-to-oval, hyperchromatic nuclei (red arrows).

Reviewer 2 Report

Comments and Suggestions for Authors

This is a well-written and clinically valuable case report addressing an extremely rare ovarian large cell neuroendocrine carcinoma (LCNEC) with integrated genomic profiling. The manuscript is clear, concise, and well-structured. The integration of histopathologic, immunohistochemical, radiologic, and genomic data adds significant novelty. The discussion is comprehensive and contextualizes the case within existing literature.
Nonetheless, some clarifications and improvements are needed to further strengthen the manuscript.

Although the rarity of pure ovarian LCNEC is emphasized, the discussion could better highlight how this case differs from or adds to the limited existing reports (e.g., comparison with Wang et al. 2024 and Flores Legarreta et al. 2024). Please specify what is unique in this patient’s genomic profile compared with published cases.

Figures 1 and 2 are informative but could benefit from improved resolution and labeling. Ensure arrows or markers clearly indicate the main features described in the text.

Consider including a panel with NGS results (mutational landscape or pathway alterations) for visual impact.

The tumor mutational burden reported (277 mutations/Mb) is extraordinarily high. This value should be discussed in more depth, including possible technical explanations or biological reasons, since such levels are unusual even in hypermutated tumors.

Please clarify whether germline BRCA2 testing was performed or excluded, please learn more about the topic on brca by referring to this paper https://doi.org/10.1159/000543869

The manuscript states no recurrence was observed over two years. Please add more details on the follow-up strategy (interval imaging, CA-125, other biomarkers).

Were any maintenance strategies (PARP inhibitors, immunotherapy) considered or discussed with the patient, given the HRD/MSI-H/TMB-H profile?

While the discussion addresses molecular convergence among gynecologic NECs, it would be useful to comment briefly on the challenges of conducting clinical trials in such rare entities and the potential role of international registries or basket trials.

Please expand on the therapeutic implications: under what conditions might PARP inhibitors or immune checkpoint inhibitors realistically be considered?

Abstract: Please avoid abbreviations (e.g., HRD, MSI-H, TMB) before first defining them.

Table 1: “Typical Ovarian LCNED” should be corrected to “LCNEC”.

References: Some citations are slightly outdated; ensure inclusion of the most recent genomic studies on ovarian NECs (2023–2024).

English style: A few sentences could be smoothed for fluency (e.g., line 227 “Establishing centralized genomic registries…” could be rephrased more concisely).

Please check consistency of gene nomenclature (e.g., italicize BRCA2, TP53, PTEN).

Round 2

Reviewer 1 Report

Comments and Suggestions for Authors

Thanks for improving the manuscript.

Comments on the Quality of English Language

NA